# Breakthrough COVID-19 Infections after Booster SARS-CoV-2 Vaccination in a Greek Cohort of People Living with HIV during the Delta and Omicron Waves [note 2]

**DOI:** 10.3390/biomedicines12071614

**Published:** 2024-07-19

**Authors:** Konstantinos Protopapas, Konstantinos Thomas, Charalampos D. Moschopoulos, Eirini Oktapoda, Eirini Marousi, Eirini Marselou, Nikiforos Stamoulis, Christos Filis, Pinelopi Kazakou, Chrysanthi Oikonomopoulou, Georgios Zampetas, Ourania Efstratiadou, Katerina Chavatza, Dimitra Kavatha, Anastasia Antoniadou, Antonios Papadopoulos

**Affiliations:** 4th Department of Internal Medicine, University General Hospital Attikon, Medical School, National and Kapodistrian University of Athens, 1 Rimini Str., Chaidari, 12462 Athens, Greece; costas_thomas@yahoo.com (K.T.); bmosxop@yahoo.gr (C.D.M.); eirini.oktapoda@hotmail.com (E.O.); marousieirini@gmail.com (E.M.); eirinimarsel@gmail.com (E.M.); stam.nikiforos@gmail.com (N.S.); christos.filis@gmail.com (C.F.); pkazakou@hotmail.com (P.K.); xrysanthi_oikonomopoulou@yahoo.gr (C.O.); zampetas@gmail.com (G.Z.); raniaefstratiadou@gmail.com (O.E.); kxavatza@gmail.com (K.C.); dimitra.kavatha@gmail.com (D.K.); ananto@med.uoa.gr (A.A.); antpapa1@otenet.gr (A.P.)

**Keywords:** HIV infection, COVID-19, breakthrough infection, vaccines, booster dose, Omicron variant

## Abstract

Introduction: Currently approved SARS-CoV-2 vaccines have been proven effective in protecting against severe COVID-19; however, they show variable efficacy against symptomatic infection and disease transmission. We studied the breakthrough COVID-19 infection (BTI) after booster vaccination against SARS-CoV-2 in people living with HIV (PWH). Methods: This was a retrospective, single-center, descriptive cohort study involving PWH, who were followed in the HIV Clinic of “Attikon” University Hospital in Athens, Greece. A BTI was defined as a case of laboratory-confirmed COVID-19 occurring at least 14 days after the third (booster) vaccine dose. Results: We studied 733 PWH [males: 89%, mean age: 45.2 ± 11.3 years, mean BMI: 26.1 ± 4.1, HIV stage at diagnosis (CDC classification): A/B/C = 80/9/11%, MSM: 72.6%] with well-controlled HIV infection. At least one comorbidity was recorded in 54% of cases. A history of ≥1 vaccination was reported by 90%, with 75% having been vaccinated with ≥3 vaccines. Four hundred and two (55%) PWH had a history of COVID-19 and 302 (41.2%) had a BTI, with only 15 (3.7%) needing hospitalization. Only one patient was admitted to the ICU, and no death was reported. Regarding BTI after booster dose, increased age (OR = 0.97, 95% CI: 0.96–0.99, per 1-year increase), and COVID-19 infection prior to booster dose (OR = 0.38, 95% CI: 0.21–0.68) were associated with a lower likelihood for BTI, whereas higher BMI (OR = 1.04, 95% CI: 1.01–1.08) and MSM as a mode of HIV transmission were associated with increased risk (OR = 2.59, 95% CI: 1.47–4.56). The incidence rate of total COVID-19 and BTI followed the epidemic curve of the general population, with the highest incidence recorded in June 2022. Conclusions: A significant proportion of PWH with well-controlled HIV infection experienced a BTI, with the majority of them having mild infection. These data, which include the period of Omicron variant predominance, confirm the importance of vaccination in the protection against severe COVID-19.

## 1. Introduction

HIV infection is characterized by immune dysregulation caused by HIV and often results in unforeseeable or uncommon clinical manifestations and worse clinical outcomes regarding coexisting infections when contrasted with people without HIV (PWoH) [1]. People with HIV (PHW) were initially identified as a high-risk group prone to experiencing exacerbated COVID-19 outcomes, including severe disease, hospitalization and mortality [2].

Consequently, the Greek vaccination program included PWH in the prioritized vulnerable groups eligible for early COVID-19 vaccination (https://emvolio.gov.gr). This initiative aligned with global trends and guidelines. This strategic measure, combined with effective information provided by healthcare providers treating PWH, translated to a high and early vaccine uptake within this population. With the advent of the Delta variant, a significant proportion of PWH became fully vaccinated [3].

Presently approved vaccines against SARS-CoV-2 have demonstrated their safety and high tolerability in the general population, a trend consistent with observations among PWH [4]. Vaccination has proven to be the most effective strategy for preventing severe COVID-19 infection. The antibody response of PWH to COVID-19 vaccines is generally satisfactory, particularly in individuals with higher CD4 cell counts and undetectable viral loads [5]. Even in cases of low CD4 counts, the cellular response appears relatively unaffected, potentially providing a degree of protection against serious infection [6]. Globally, data indicate the varying efficacy of currently available vaccines against symptomatic infection in immunocompromised individuals, a trend accentuated after the emergence of the Omicron variants [7].

While breakthrough COVID-19 infections (BTIs) have been extensively described in the general population, relatively limited amounts of data exist concerning BTI in PWH, especially during the Omicron period. Existing knowledge points to elevated rates of BTIs in immunocompromised patients, fully vaccinated PWH included, when compared to the general population; however, this evidence derives mainly from studies conducted during the pre-Omicron period [8,9]. This study aims to estimate the incidence, severity and risk factors associated with BTIs subsequent to booster SARS-CoV-2 vaccination in a PWH cohort during the periods of Delta and Omicron variant predominance.

## 2. Materials and Methods

### 2.1. Patients

We conducted a retrospective, single-center, cohort study considering PWH followed at the HIV outpatient Clinic of “Attikon” University Hospital, Athens, Greece. The study period was from March 2020 until the end of October 2022.

After obtaining written informed consent from the participants, we performed chart reviews and personal interviews and data-gathering regarding demographics, HIV infection and comorbidities, and the histories of vaccination and COVID-19 incidence were recorded. We have used the United States Center for Disease Control and Prevention (CDC) classification of HIV infection (category A, asymptomatic PWH; category B, those not in category A or C; and category C, AIDS-defining illness). None of the included patients were included in the study during acute HIV infection. The participants were interviewed by the staff of the HIV Clinic (physicians and a nurse). COVID-19 cases were defined as individuals exhibiting symptoms of acute respiratory illness (ARI) and testing positive through PCR or antigen tests of upper or lower respiratory tract samples. For cases tested in health facilities rather than at-home self-testing, positive test dates were cross-verified with the National COVID-19 Registry, which maintains records of all PCR and laboratory-performed antigen tests. Data regarding vaccination dates and administered vaccine types were retrieved from the national COVID-19 vaccination registry. The primary outcome of the study was the incidence of laboratory-confirmed breakthrough COVID-19 infection in participants at least 14 days after the 3rd vaccine (booster) dose. The periods of Delta-variant predominance and Omicron predominance were defined based on nationwide data from the National Public Health Organization (NPHO), defining periods spanning from 1 August 2021 to 20 January 2022 and from 21 January 2022 onward until the end of the follow-up (31 October 2022), respectively (NPHO, https://eody.gov.gr/ and personal communication [10]). The study was approved by the Institutional Ethics Committee of Attikon University Hospital (ID: 487/3-9-2020).

### 2.2. Statistical Analysis

Dichotomous variables are presented as percentages, and continuous variables as means (standard deviation, SD) for normal and medians (interquartile range, IQR) for nonparametric distributions, respectively. A chi-squared test was used to compare dichotomous variables, and a *t*-test for continuous variables. A multivariate Cox regression analysis with time-varying covariate (4th vaccine dose) was performed to identify predictors of BTI after the booster vaccine dose. Variables included in the statistical model were those of biologic significance (age and sex) and those with a *p*-value of <0.1 in univariate analysis. The cumulative incidence of BTI was calculated with Kaplan–Meier analysis and a log-rank test was implemented for the comparison of the BTI incidence among subgroups. The incidence of COVID-19 infections and BTIs are presented as cases per 10,000 patient-days, with their respective 95% confidence intervals (95% CI). Statistical analyses were performed with the R project for Statistical Computing (version 4.3.2), SPSS (IBM SPSS Statistics for Windows, v. 25.0. Armonk, NY, USA: IBM Corp) and OpenEpi. The threshold of statistical significance was set as a *p* value < 0.05 for all comparisons.

## 3. Results

### 3.1. Patients’ Characteristics

Seven hundred and thirty-three PWH followed in our HIV Unit were studied. Six hundred and fifty-one (89.1%) were males, with a mean age of 45.2 ± 11.3 years and 639 (93.4%) were of Greek nationality. Regarding HIV infection, 80.3%, 8.6% and 11.1% had been diagnosed with stage A, B and C, respectively (CDC classification). Men having sex with men (MSM) was the most common mode of transmission (72.6%), followed by heterosexual transmission (21.2%), while people who injected drugs (PWID) represented 5.2% of the PWH. During the most recent follow-up, the majority of patients had a CD4 count of >500 c/μL (80.9%), with only 8.4% having <350 c/μL. All but nine patients (99.3%) were on antiretroviral therapy. More than half of the participants (n = 394, 54.3%) reported at least one comorbidity, with active smoking being the most common (52.1%), followed by dyslipidemia (30.7%), arterial hypertension (17.3%) and obesity (BMI ≥ 30 kg/m^2^, 14%). A small proportion of PWH were on immunosuppressive therapy for chronic inflammatory diseases or cancer chemotherapy (5.9 and 1.9%, respectively). The patients’ characteristics are presented in Table 1 in more detail.

### 3.2. Vaccination against SARS-CoV-2, COVID-19 Infections, and Breakthrough Infections (BTIs) after the Booster Vaccine Dose

Almost nine out of ten patients (n = 657, 89.6%) had received ≥1 dose of vaccine against SARS-CoV-2, with 550 (75.1%) reporting vaccination with ≥3 doses and the vast majority of the administered vaccines being mRNA-based (78.9–100%, depending on the dose) (Table 1). PWH with a history of three doses of vaccination were older (46 ± 11.2 vs. 42.4 ± 11 years, *p* < 0.001) and more likely to be males (92.2% vs. 79.8%, *p* < 0.001) and to have arterial hypertension (19.3% vs. 11.5%, *p* = 0.016), while being less likely to have viral hepatitis (5.3% vs. 9.8%, *p* = 0.03). Compared to non-PWID cases, the proportion of PWID patients with a history of at least three vaccine doses was lower, although this was not a statistically significant difference (62.2% vs. 75.6%). The administration of the first and second doses peaked during April 2021 and May 2021, respectively. A second, lower, peak in the administration of the first dose occurred during the Delta variant’s predominance in November 2021. Most of booster doses were administered in December 2021. None of them were bivalent mRNA boosters, as that type of vaccine was introduced during the second half of 2022 (https://emvolio.gov.gr/). The temporal trends associated with vaccination with the first three doses are shown in Appendix A.

During the study period, 402 PWH (54.9%) developed COVID-19. Among those with ≥3 vaccine doses, 202 developed a BTI after the third (booster) dose (40.4%) (Table 1). Of note, the epidemic curves for the first COVID-19 infection and first BTI after the booster dose mirrored those for the general population in Greece, with the highest incidence rates observed in June 2022 (during Omicron-variant predominance) for both the first COVID-19 infection (33.61/10,000 patient-days, 95% CI: 25.06–44.19) and for the first BTI after the booster dose (33.83/10,000 patient-days, 95% CI: 24.6–45.4) (Figure 2A,B). We sought to estimate the incidence of BTI according to the time elapsed after the booster dose. No significant difference in the BTI incidence rates was found after the first 90 days or for the period between vaccination and 90 days since vaccine administration (19.1 vs. 16.7 per 10,000 patient-days, OR = 1.14, 95% CI: 0.84–1.53).

We then focused particularly on PWH vaccinated with ≥3 doses. Among them, 220 (40.4%) developed a BTI after the booster dose, during follow-up. Compared to those who did not develop a BTI after the booster dose, these participants were younger (44.2 ± 10.2 vs. 47.4 ± 12 years, *p* = 0.001), had lower prevalence of arterial hypertension (14.5% vs. 22.5%, *p* = 0.02), had a trend for higher BMI (26.2 ± 4.2 vs. 25.5 ± 4.2 kg/m^2^, *p* = 0.10) and were less likely to have a prior history of COVID-19 before the booster dose (5.9% vs. 17.6%, *p* < 0.001) or to have received additional vaccine doses after the booster (8.2% vs. 16.4%, *p* = 0.005) (Table 2). In multivariable analysis, older age (per 1-year increase OR = 0.97, 95% CI: 0.96–0.99) and prior COVID-19 infection (OR = 0.38, 95% CI: 0.21–0.68) were associated with lower risk for BTI, whereas higher BMI (OR = 1.04, 95% CI: 1.01–1.08) and MSM (OR: 2.59, 95% CI: 1.47–4.56) with higher risk **(**Table 3, Figure 1). The cumulative incidence of BTI after the booster dose in relation to the age and history of prior COVID-19 infection are shown in Figure 2A,B.

### 3.3. Hospitalizations Due to COVID-19 Infection

Out of a total of 402 COVID-19 cases, only 15 (3.7%) were hospitalized. Thirteen (87%) of those cases occurred during the pre-Omicron period, with all but one of the patients being non-vaccinated. Two patients needed hospitalization during the period of Omicron-variant predominance; one was non-vaccinated, and one had received two vaccine doses before admission. Only one (6.7%) patient needed ICU admission, and that patient survived; no deaths were reported. Compared to those not hospitalized, these 15 patients were older (52.1 ± 12.7 vs. 42.9 ± 10.7 years, *p* = 0.001) and more likely to have been diagnosed with HIV stage C (33.3% vs. 8.8%, *p* = 0.005) and to have chronic viral hepatitis (20% vs. 3.9%, *p* = 0.003).

## 4. Discussion

In this study, we present a detailed description of COVID-19 BTI in our cohort of Greek PWH. Our main findings include the following: Firstly, we found a significant adherence to vaccination guidelines, with three out of four PWH having been vaccinated with at least three doses. Secondly, we demonstrated that the epidemic curves of total COVID-19 infections and BTI followed the respective curves of the general population. Thirdly, we identified several variables associated with BTI in PWH vaccinated with at least three doses. Finally, we showed that, despite the high incidence of BTI, the burden of severe COVID-19 in this population was low and particularly pertained to unvaccinated persons, with the vast majority of cases being mild.

Among PWH of our cohort, only 12.2% had received no dose of vaccine or only one dose, whereas 75% had received at least three doses. In accordance with other studies referencing PWH, we found increased vaccine coverage to be associated with patients with higher ages and male PWH [11,12]. Interestingly, other studies have shown similar vaccine coverage patterns in PWID, along with a positive association between HIV seropositivity and SARS-CoV-2 vaccine uptake [13]. Due to the homogenous composition of our cohort in terms of racial/ethnic groups, we were not able to assess the impact of this factor on vaccine uptake.

The epidemic curves of total COVID-19 infections and BTI followed the respective ones of the general population, peaking in June 2022. The reported peak incidence here does not differ from the rates respectively described by others during the same period of Omicron predominance [14]. This could be explained by the increased immune escape associated with the Omicron variant in individuals vaccinated with the initially approved vaccines [15], as well as the rapid waning of vaccine effectiveness against this variant [16]. Those monovalent vaccines did not efficiently interrupt SARS-CoV-2 transmission but were quite effective at preventing severe disease [17].

Several studies have attempted to describe risk factors for BTI in the general population. Among them, lower educational level, younger age, primary vaccination with mRNA-based vaccines and crowding in indoor spaces have been associated with an increased risk for BTI [18]. As for severe BTI, older age, male sex, higher comorbidity index and specific comorbidities have been linked with increased risk [19]. Booster vaccine doses have profoundly protective and long-lasting effects on the risk for severe BTI, even for Omicron variant [19,20,21]. HIV infection was found to confer a 30% increase in the risk for severe BTI, lower than seen with other conditions, such as iatrogenic immunosuppression [22]. Lang et al. reported a low rate of severe BTI among vaccinated PWH, which was similar to the respective rate of people without HIV. Female sex, older age, cancer diagnosis and low CD4 count were associated with severe COVID-19 BTI, whereas previous COVID-19 was found to be protective against severe BTI [23]. Rasmussen et al. performed a nationwide, population-based study in Denmark focusing on laboratory-confirmed infections, irrespective of BTI status. They showed that among PWH, an additional third vaccine dose resulted in a 10% reduction in BTI, an 80% reduction in the risk of death for individuals over 60 years old and a 40% reduction in the risk of hospitalization [24]. In a study that included the pre-Delta and Delta periods, Coburn et al. reported a low incidence of BTI for up to 9 months post-full-vaccination, both in PWH and in people without HIV. Younger age, prior COVID-19 before full vaccination, not having received additional vaccine doses and CD4 counts below 500 cells/mm^3^ were independently associated with BTI [8]. In a recent study by Yang et al. which was conducted during the pre-Omicron and Omicron circulation periods and included both PWH and PWoH, BTI risk was lower in those with prior COVID-19 before full vaccination and those having received a booster dose, whereas the BTI risk was increased during the Omicron period. In the subset of PWH, no statistical significant association between CD4 count and BTI was reported.

Some of our findings are in accordance with those of the aforementioned studies. Increased age and a history of COVID-19 prior to the booster dose were independently associated with a lower likelihood for BTI, whereas higher BMI and MSM as a mode of transmission were associated with increased risk. We reaffirmed the findings of Tseng et al. [17] showing the substantial effectiveness of vaccination against severe COVID-19, since only three out of fifteen infections needing hospitalization occurred in vaccinated PWH [17]. While increased age is correlated with severe BTI and, similar to Coburn et al. [8] and studies from people without HIV [18], younger PWH were more likely to develop non-severe BTI in our cohort, this is a finding possibly explained by the increased participation of people from those age groups in social gatherings or by the stricter adoption of social distancing and protective measures by older PWH with comorbidities. It has to be noted, however, that our population was younger than those included in the aforementioned studies. For example, in the study by Coburn et al., 67% of PWH were older than 55 years old, whereas the respective percentage in our study was 21.1%. Similarly, the proportion of PWH over 60 years old in the study of Yang et al. was 28% [25], with the respective rate in our study being 11%. In our study, an additional booster (fourth) dose was not shown to reduce the risk for COVID-19 BTI [26]. This finding comes in contrast with those from Lau et al., who reported a 35% decrease in the general population 100 days after the fourth mRNA-based vaccine dose [16], as well as the respective decrease seen in the study by Coburn et al. after a third mRNA vaccine dose in PWH (29–50%) [8]. Yang et al. also reported a 79% decreased risk for BTI with an additional third dose [25]. Our results are, however, in concordance with the accumulated data regarding the vaccine efficacy of a fourth vaccine dose against Omicron infection, showing a rapidly attenuating protection after the first 90–100 days [27,28].

Prior COVID-19 seems to provide an additional protection against subsequent reinfection, with its protective effect being significantly less pronounced for Omicron variants [29,30]. We found a 62% decrease in the likelihood for BTI in PWH with a history of prior COVID-19, one similar to the 63% decrease in BTI reported in a recent study of PWoH and PWH by Yang et al. [25]. Interestingly, Lang et al. focused on severe BTI and reported a 79% reduction in cases in which the subjects had a history of COVID-19 before full vaccination [23]. We also found a 4% increase in the risk for BTI for each 1-point increase in BMI. Recent data showed that obesity not only increases the risk for severe COVID-19 outcomes in young people [31,32], but also may also impair the vaccines’ effectiveness [33]. Finally, we identified MSM as a distinct risk factor for BTI after three vaccine doses. This could be explained by a rebound in public-event attendance or high-risk sexual behavior that might have facilitated viral transmission, thus leading to a disproportionate increase in breakthrough COVID-19 infections in this group of PWH. A recent study from China showed rebounds in the prevalence of cases with multiple sex partners, mobility for sexual activity and the use of recreational substances post-pandemic, although to lower levels than found in the pre-pandemic period [34]. This change in sexual activity patterns also is reflected in the increase in the incidence of sexually transmitted infections after the pandemic, with which COVID-19 shares some common routes of transmission [35]. To our knowledge, this is the first study that recognizes increased BMI and MSM as risk factors for BTI in PWH. CD4 count was not found to be associated with BTI; however, this could be explained by the low proportion of PWH with <350 cells/mL (8.4%).

We believe that there are several strengths associated with our study. First, it included the Omicron-predominance period, something that was absent from previous studies including PWH. In addition, we collected and analyzed patient-level data, thus reducing the risk for non-identified confounding factors. Moreover, and in contrast to other studies reporting the characteristics of BTI after the initial two vaccine doses, we focused on breakthrough infections after an additional third booster dose. Given this, we believe that our study is more relevant and closer to the real-world conditions of COVID-19 vaccination, within which most patients with risk factors have received at least one booster dose. Finally, although this is a single-center study, we managed to include almost 75% of the PWH followed in our unit, abbreviating, but not eliminating, the risk for selection bias. This study also comes with several limitations. Since this is a single-center study, our data cannot be generalized to the entire Greek HIV population, nevertheless more than 10% of the total Greek PWH were followed in our unit. Moreover, the most vulnerable PWH, who are not optimally linked to care (e.g., immigrants and people who inject drugs), were less likely to be included. Data regarding socioeconomic status, a well-defined variable associated with vaccine uptake and outcomes of COVID-19 [36], were not analyzed. In addition to this, the presence of other non-identified health conditions that could affect the outcome cannot be excluded. We also acknowledge that our study population was mainly young, male PWH and for this reason our findings may not be easily extrapolated to populations with more elderly or female patients. The absence of a comparator group (PWoH) constitutes another limitation of our study, one that precludes the generalization of our results regarding the incidence of BTI to the general population. Nevertheless, a recent study which compared the BTI rates between PWoH and PWH with high CD4 counts (66% with >500 cells/mm^3^) did not find any difference [25]. Finally, as the pandemic progressed, home self-testing gained more ground. Since these results were not registered in the National COVID-19 Registry, self-reports of positive tests were considered to be sufficient for confirmed infections. Consequently, there is a possibility that the actual number of infections has been under- or over-estimated.

## 5. Conclusions

In conclusion, although COVID-19 BTI were frequent in our PWH population, disease severity was mild and adverse outcomes were rare. These data further emphasize the significance of adequate vaccination against SARS-CoV-2.

## Figures and Tables

**Figure 1 biomedicines-12-01614-f001:**
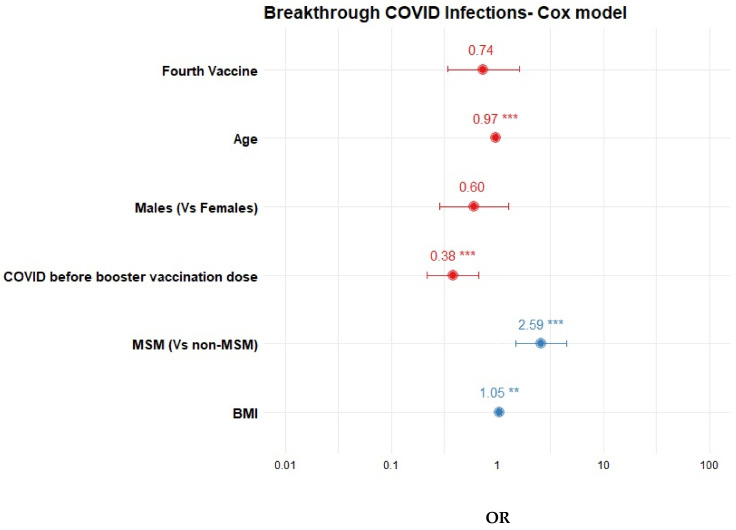
Forest plot of multivariate Cox regression analysis of factors associated with BTI after booster vaccine dose. Note: Odds ratios (ORs) lower than 1 are depicted in red and ORs higher than 1 in blue. *p* values <0.01 are marked with (**) and those <0.001 with (***).

**Figure 2 biomedicines-12-01614-f002:**
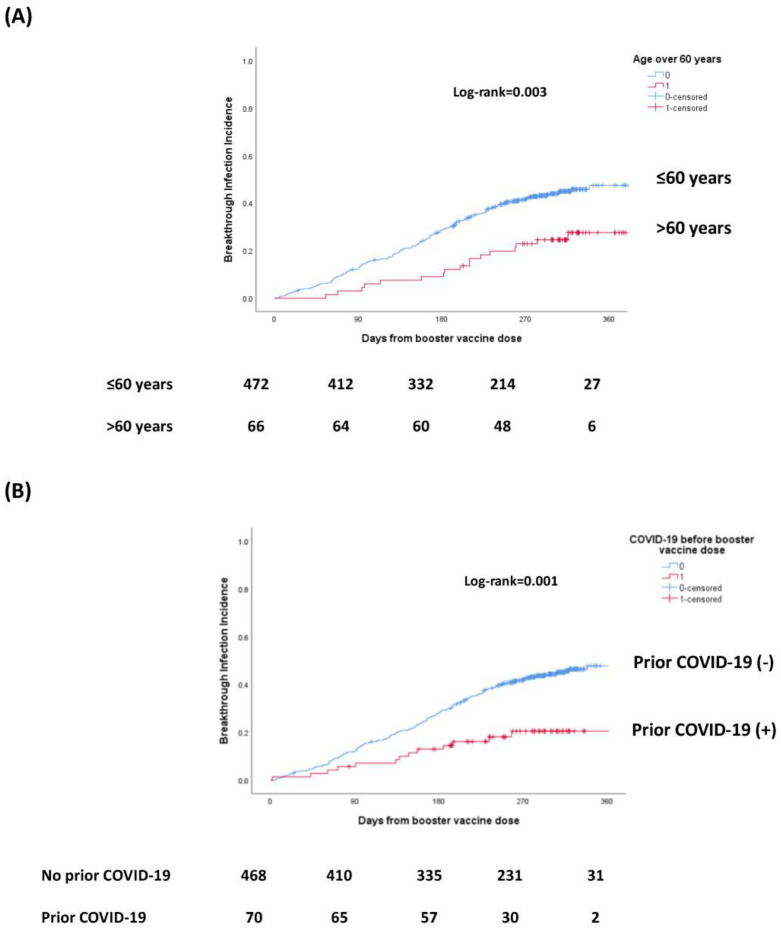
Kaplan–Meier curves, with the respective numbers of persons at risk showing the cumulative incidence of BTI after the 3rd vaccine dose according to the age (**A**) and history of prior COVID-19 infection (**B**).

**Table 1 biomedicines-12-01614-t001:** Descriptions of the PWH participating in the study (n = 733).

Variable	Number	Results
Male, n (%)	731	651 (89.1)
Age, years, mean (SD)	733	45.2 (11.3)
Age >60 years, n (%)	733	82 (11.2)
Greek nationality, n (%)	684	639 (93.4)
ART, n (%)	729	724 (99.3)
Mode of transmission, n (%)	716	
MSM		520 (72.6)
PWID		37 (5.2)
Heterosexual		152 (21.2)
Transfusion		7 (1.0)
HIV category (CDC classification), n (%)	731	
A		587 (80.3)
B		63 (8.6)
C		81 (11.1)
Last CD4 count (c/μL), mean (SD)	732	787 (345)
Nadir CD4 count (c/μL), mean (SD)	734	316 (203)
CD4 count (c/μL)	728	
<200		13 (1.8)
200–349		48 (6.6)
350–499		78 (10.7)
>500		589 (80.9)
≥1 comorbidity, n (%)	725	394 (54.3)
COPD, n (%)	733	30 (4.1)
Active smoking, n (%)	733	382 (52.1)
BMI (kg/m^2^), mean (SD)	681	26.1 (4.1)
Asthma, n (%)	733	52 (7.1)
Diabetes, n (%)	733	35 (4.8)
Coronary artery disease, n (%)	733	20 (2.7)
Hypertension, n (%)	733	127 (17.3)
Dyslipidemia, n (%)	733	225 (30.7)
Autoimmune diseases, n (%)	733	30 (4.1)
Chronic viral hepatitis, n (%)	733	47 (6.4)
Immunosuppressive therapy, n (%)	660	39 (5.9)
Chemotherapy, n (%)	733	14 (1.9)
≥1 vaccine, n (%)	733	657 (89.6)
Number of vaccines, n (%)	733	
0		76 (10.4)
1		13 (1.8)
2		94 (12.8)
3		471 (64.3)
4		71 (9.7)
5		8 (1.1)
mRNA-based vaccines, n (%)		
1st dose	650	513 (78.9%)
2nd dose	635	529 (83.3%)
3rd dose	544	537 (98.7%)
≥4th dose	78	78 (100%)
COVID-19, n (%)	733	402 (54.9)
Breakthrough COVID-19 *, n (%)	657	302 (46)
Breakthrough COVID-19 after booster dose **, n (%)	500	202 (40.4)
CD4 count at COVID-19 infection (c/μL) ***, mean (SD)	402	787 (346)
Hospitalization ***, n (%)	402	15 (3.7)
ICU admission ***, n (%)	402	1 (0.2)

* Among n = 657 who received ≥1 vaccine dose. ** Among n = 500 who received a booster dose and had not experienced a prior breakthrough infection. *** Among n = 402 with a history of COVID-19 infection.

**Table 2 biomedicines-12-01614-t002:** Comparisons of PWH who developed COVID-19 BTI after the booster dose vs. those who did not.

	COVID-19 BTI (−) n = 324	COVID-19 BTI (+) n = 220	*p*
Male, n (%)	293 (91%)	208 (94.4%)	0.12
Age, years, mean (SD)	47.4 (12.0)	44.2 (10.2)	0.001
Age group, n (%)			0.05
20–29	13 (4)	12 (5.5)	
30–39	76 (23.5)	61 (27.7)	
40–49	109 (33.6)	89 (40.5)	
50–59	76 (23.5)	41 (18.6)	
60–69	33 (10.2)	12 (5.5)	
≥70	17 (5.2)	5 (2.3)	
Greek nationality, n (%)	290 (94.2%)	193 (97%)	0.14
Category A (CDC classification) at diagnosis, n (%)	251 (77.7%)	177 (80.8%)	0.49
MSM, n (%)	224 (70.2%)	182 (86.3%)	<0.001
CD4 count at booster dose (c/μL), median (IQR)	722 (528–963)	749 (590–972)	0.12
Detectable viral load, n (%)	16 (5%)	18 (8.2%)	0.13
Additional vaccine dose(s), n (%)	53 (16.4%)	18 (8.2%)	0.005
COPD, n (%)	13 (4%)	10 (4.5%)	0.79
Smoking, n (%)	171 (52.8%)	104 (47.3%)	0.49
BMI	25.5 (4.2)	26.2 (4.2)	0.10
Diabetes, n (%)	19 (5.9%)	8 (3.6%)	0.24
Hypertension, n (%)	73 (22.5%)	32 (14.5%)	0.02
Dyslipidemia, n (%)	110 (34%)	61 (27.7%)	0.12
Immunosuppression or chemo, n (%)	18 (5.6%)	13 (5.9%)	0.86
Prior COVID-19 before booster dose, n (%)	57 (17.6%)	13 (5.9%)	<0.001

**Table 3 biomedicines-12-01614-t003:** Multivariate logistic regression analysis for predictors of COVID-19 BTI after the booster vaccine dose.

Variable	OR	95% CI	*p*
Age (per 1-year increase)	0.97	0.96–0.99	0.0001
Male sex (vs. female)	0.60	0.28–1.31	0.20
MSM (vs. non-MSM)	2.59	1.47–4.56	0.0009
Additional 4th vaccine dose	0.73	0.36–1.49	0.39
COVID-19 before booster dose	0.38	0.21–0.68	0.0009
BMI	1.04	1.01–1.08	0.009

## Data Availability

Patient-level data would be available to interested researchers upon reasonable request.

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
