# Peer review of "Breakthrough COVID-19 Infections after Booster SARS-CoV-2 Vaccination in a Greek Cohort of People Living with HIV during the Delta and Omicron Wavesâ€"

_biomedicines, 2024, doi:10.3390/biomedicines12071614_

Round 1
Reviewer 1 Report
Comments and Suggestions for Authors
Report – Biomedicines (MDPI)
Received: 22 May 2024
Submission ID: biomedicines-3044021
Title: Breakthrough COVID-19 infections after booster SARS-CoV-2 vaccination
in a Greek cohort of people living with HIV during Delta and Omicron wave - a
single center experience.
Authors: Konstantinos Protopapas *, Konstantinos Thomas, Charalampos D
Moschopoulos, Eirini Oktapoda, Eirini Marousi, Eirini Marselou, Nikiforos
Stamoulis, Christos Filis, Pinelopi Kaakou, Chrysanthi Oikonomopoulou,
Georgios Zampetas, Ourania Efstratiadou, Katerina Chavatza, Dimitra Kavatha,
Anastasia Antoniadou, Antonios Papadopoulos.
Comments to the authors:
Summary of the study:
In this retrospective, single-center, cohort study on people living with HIV (PLH) at the HIV outpatient Clinic of “Attikon” University Hospital (Athens, Greece), the authors studied COVID-19 breakthrough infections after the patients received mostly mRNA-based vaccines. The study period included the timespan when the Delta and Omicron variants predominated in Greece. The PLH were mostly males of Greek nationality, with acute HIV infection and on cART and with CD4 T-cell counts >500 cells/mm3. Just over half of the PWH had comorbidities, that included active smoking, dyslipidemia and hypertension. The vast majority received at least one COVID-19 vaccine dose. Four-hundred-and-two PWH acquired COVID-19 infection, of which 302 acquired breakthrough infection after the initial COVID-19 immunizations and 202 acquired breakthrough infection after the booster vaccination. The authors found significant associations between increased risk of breakthrough infection with younger age, a higher BMI and MSM as mode of infection, with a reduced risk of breakthrough infection in those that were infected with COVID-19 prior to the booster vaccination or those that received a greater number of booster vaccinations. Only 15 of the PWH were hospitalized, with the majority of them not being vaccinated. In conclusion, the authors have identified several variables associated with COVID-19 breakthrough infections in PWH at their study site and found that the burden of severe COVID-19 at their study site was low.
Major concerns with the study:
1. Page 3, Table 1 (under “HIV stage, n (%)”)
Comment: I find it peculiar that so many (80.3%) of the PWH were in the acute stage (i.e., stage A - within 2 to 4 weeks after infection with HIV) of HIV infection. A recent WHO report on the in HIV/AIDS surveillance in Europe (2021 – 2020 data) reported that only 8.4% of cases were diagnosed during acute infection (irrespective of transmission route), with only 13.9% of cases reported as acute infections in MSM.
(Ref.:https://www.ecdc.europa.eu/sites/default/files/documents/HIV-AIDS_surveillance_in_Europe_2023_%28_2022_data_%29_0.pdf)
a. Can the authors please clarify if they are referring to acute HIV infection and/or recent HIV infection when they refer to “HIV stage A” in Table 3.
b. If they are referring to acute HIV infection, can the authors please
i. state their definition of acute HIV infection (i.e., HIV stage A) in the text
ii. explain how/why so many acute HIV infections were captured in their study site (i.e., was there a “test-and-treat” campaign that was happening at the time of the study, etc.?)
iii. why 99.3% of the PWH were on cART? Surely, the 80.3% of acutely infected PWH would not have been on cART?
c. Please note on the use of wording in the abstract (lines 25 -26: “HIV stage at diagnosis: A/B/C=80/9/11%, MSM: 72.6%) with 25 well-controlled HIV infection.”), which would not be true if the majority of patients were in acute infection. Can the authors please reword those lines?
2. Page 3, Table 1 (regarding the numbers and percentages)
Comment 1: The percentages in the table are not correct (in most cases) if n=733 is used as the denominator. Can the authors please make sure of the calculations to obtain the percentages, and correct it in the tables and the corresponding text?
Comment 2: Under “mRNA based vaccines, n (%)”
i. The numbers are not shown, only the percentages. Can the authors please also include the numbers in the table?
ii. The percentages for the different doses look like they were inverted: 1st dose (78.9%), 2nd dose (83.3%), 3rd dose (98.7%), ³4th (100%)… should be : 1st dose (100%), 2nd dose (98.7%%), 3rd dose (83.3%), ³4th (78.9%)?? Can the authors please correct this in the table and corresponding text?
3. Page 3, Table 1 (regarding “Number of vaccines, n (%)” vs. “mRNA based vaccines, n (%)”
Comment: The percentages under “mRNA based vaccines, n (%)” do not match with the numbers under “Number of vaccines, n (%)”. For example,
Number of vaccines: (n=0)76 + (n=1)13 + (n=2)94 + (n=3)466 + (n=4)70 + (n=5)8 = 727 (not 733)
Thus, for mRNA vaccines ≥4th dose: (n=4)70 + (n=5)8 = 78 total
For total number of vaccines: 727 – 76(not vaccine) = 651 total
For 4 + 5 vaccines = 70 + 8 = 78/651 = 11.98%
However, under “mRNA based vaccines, n (%)”, the percent for ≥4th dose is 100% (78.9% with reference to my earlier comment).
Can the authors please make sure of the calculations to obtain the percentages, and correct it in the tables and the corresponding text?
4. Page 9, lines 215 – 234
Comment: The effect of age on BTI seems to have variable associations (positive and negative) based on the literature that the authors have referred to and may be related to different age groups that were studied in the current studies. Can the authors please highlight this in their discussion?
5. Page 7, Figure 2A: regarding age
Comment: The Kaplan-Meier curves are shown for PWH under and over 60 years old. However, according to Table 2, taking that average and standard deviations inconsideration, none of the PWH were 60 or older than 60. It is unclear which data is used for the figure. Can the authors please clarify this in the text?
6. Page 7, Figure 2B: regarding prior COVID-19
Comment: Lines 158 – 160 (page 5) states that “In multivariable analysis, older age (per 1-year increase OR=0.97, 95% CI: 0.96- 158 0.99) and prior COVID-19 infection (OR=0.38, 95% CI: 0.21-0.68) were associated with lower risk for BTI…”. However, Figure 2B shows that incidence for breakthrough infections is high in younger (<60) PWH. It seems that the curves could have been labelled incorrectly. Can the authors please correct this? However, if the labels are correct, can the author please provide the source of the data for this figure, and discuss how this relates to their data and findings?
7. Page 5, lines 151 – 163 and Table 2 & 3: “Compared to those who 152 did not develop BTI after the booster dose, these participants were younger (44.2 ± 10.2 vs 153 47.4 ± 12 years, p=0.001)…” and “In multivariable analysis, older age (per 1-year increase OR=0.97, 95% CI: 0.96- 158 0.99) and prior COVID-19 infection (OR=0.38, 95% CI: 0.21-0.68) were associated with lower risk for BTI,…”
Comment: The age ranges between the BTI and no-BTI seem very close (44.2 vs. 47.4) but were associated with lower risk for BTI. In lines 215 – 234 (9), studies are cited that refer to “younger age” and “older age”. How “young” or “old” were these references referring to? Since difference in age is one of the main findings of this study, it needs to be presented better, especially since both the “younger” and “older” groups in this study are on average in the mid-40’s. Can the authors please present a figure or table that indicates the number of BTI and non-BTI per age group in 2 or 5 year intervals? This will clarify the “spread” in numbers of PWH in age intervals in the non-BTI and BTI to highlight the differences in age.
Minor concerns with the study:
1. Page 2, line 72 - 73: “The participants were interviewed by the staff of the HIV Clinic (physicians and nurse).”
Suggested change: “The participants were interviewed by the staff of the HIV Clinic (physicians and a nurse).” Or “and nurses”.
2. Page 2, line 86: “COVID-19 infection in participants at least 14 days after the 3rd vaccine (booster) dose..“
Suggested change: “COVID-19 infection in participants at least 14 days after the 3rd vaccine (booster) dose.“
3. Page 6, Table 2: the text in the column headers are currently not visible. Please correct.
4. Page 7, Figure 1 legend: “Figure 1. Forest plot of multivariate Cox regression analysis of factors associated with BTI after 170 booster vaccine dose. Note: Odds ratios (ORs) lower than 1 are depicted in red and ORs higher than 1 in 171 blue. P values <0.001 are marked with (**) and those <0.001 with (***).”
Comment: ** and *** both refer to p<0.001 and p<0.001. Please correct.
5. Page 8, lines 209 – 210: “not differ from the respective rates described by others during the same period of Omin-cron predominance.”
Suggested change: “not differ from the respective rates described by others during the same period of Omni-cron predominance.”
6. Page 9, line 238: “transmission were associated with increased risk. We reaffirmed the substantial effective”
7. Suggested change: “transmission were associated with increased risk. We reaffirmed the findings of Tseng et al. that substantial effective”
8. Page 10, lines 273 – 274: In addition, we collected and analyzed patient-level data, thus reducing the risk for noni-dentified confounding factors”
Suggested change: “Page 10, lines 273 – 274: In addition, we collected and analyzed patient-level data, thus reducing the risk for non-identified confounding factors”
Author Response
Major concerns with the study:
- Page 3, Table 1 (under “HIV stage, n (%)”)
Comment: I find it peculiar that so many (80.3%) of the PWH were in the acute stage (i.e., stage A - within 2 to 4 weeks after infection with HIV) of HIV infection. A recent WHO report on the in HIV/AIDS surveillance in Europe (2021 – 2020 data) reported that only 8.4% of cases were diagnosed during acute infection (irrespective of transmission route), with only 13.9% of cases reported as acute infections in MSM.
(Ref.:https://www.ecdc.europa.eu/sites/default/files/documents/HIV-AIDS_surveillance_in_Europe_2023_%28_2022_data_%29_0.pdf)
- Can the authors please clarify if they are referring to acute HIV infection and/or recent HIV infection when they refer to “HIV stage A” in Table 3.
- If they are referring to acute HIV infection, can the authors please
- state their definition of acute HIV infection (i.e., HIV stage A) in the text
- explain how/why so many acute HIV infections were captured in their study site (i.e., was there a “test-and-treat” campaign that was happening at the time of the study, etc.?)
iii. why 99.3% of the PWH were on cART? Surely, the 80.3% of acutely infected PWH would not have been on cART?
- Please note on the use of wording in the abstract (lines 25 -26: “HIV stage at diagnosis: A/B/C=80/9/11%, MSM: 72.6%) with 25 well-controlled HIV infection.”), which would not be true if the majority of patients were in acute infection. Can the authors please reword those lines?
We have used the United States Center for Disease Control and Prevention (CDC) classification of HIV infection and not the Fiebig classification. Given this, the stage A refers to asymptomatic people living with HIV, category B refers to those not in category A or C and category C to those with AIDS-defining illness. In order to avoid misinterpretation, we revised “stage” to “category” and we clarify that we used the CDC HIV classification both in abstract (line 59), Methods (lines 107-111), Table 1 (page 6) and Table 2 (page 9). We finally report in the Methods that none of the included patients were included in the study during acute infection (lines 109-110). The inclusion of patients without acute HIV infection justifies the high proportion of cART prescription according to all current guidelines.
- Page 3, Table 1 (regarding the numbers and percentages)
Comment 1: The percentages in the table are not correct (in most cases) if n=733 is used as the denominator. Can the authors please make sure of the calculations to obtain the percentages, and correct it in the tables and the corresponding text?
Thank you for this comment. The calculations are correct, however the denominator (valid non-missing data) is different for each variable and not always n=733. We have revised Table 1 by adding a column where we report the valid numbers (denominators). We also added footnotes in Table 1, showing the variables where the denominator is by default not equal to the number of patients in the total cohort (i.e. breakthrough COVID-19 among those vaccinated).
Comment 2: Under “mRNA based vaccines, n (%)”
- The numbers are not shown, only the percentages. Can the authors please also include the numbers in the table?
Thank you, we included the valid (available) numbers for each dose, as well as the result in absolute numbers (Table 1).
- The percentages for the different doses look like they were inverted: 1st dose (78.9%), 2nd dose (83.3%), 3rd dose (98.7%), ³4th (100%)… should be : 1st dose (100%), 2nd dose (98.7%%), 3rd dose (83.3%), 4th (78.9%)?? Can the authors please correct this in the table and corresponding text?
Thank you for your comment. We here report the percentage of vaccines that were mRNA-based and not viral vector vaccines. The increase in the proportion of mRNA based vaccine is expected, since the Greek national COVID-19 vaccination program gradually moved from mixed mRNA and viral vector vaccines to the almost exclusive use of mRNA vaccines.
- Page 3, Table 1 (regarding “Number of vaccines, n (%)” vs. “mRNA based vaccines, n (%)”
Comment: The percentages under “mRNA based vaccines, n (%)” do not match with the numbers under “Number of vaccines, n (%)”. For example,
Number of vaccines: (n=0)76 + (n=1)13 + (n=2)94 + (n=3)466 + (n=4)70 + (n=5)8 = 727 (not 733)
Thus, for mRNA vaccines ≥4th dose: (n=4)70 + (n=5)8 = 78 total
For total number of vaccines: 727 – 76(not vaccine) = 651 total
For 4 + 5 vaccines = 70 + 8 = 78/651 = 11.98%
However, under “mRNA based vaccines, n (%)”, the percent for ≥4th dose is 100% (78.9% with reference to my earlier comment).
Can the authors please make sure of the calculations to obtain the percentages, and correct it in the tables and the corresponding text?
Thank you for the comment, we acknowledge that there was an error in the number of vaccines, however not substantially affecting the results. The revised numbers in Table 1 are as follows: 0=76 (10.4), 1=13 (1.8), 2=94 (12.8), 3=471 (64.3), 4=71 (9.7), 5=8 (1.1)
Accordingly, we revised the Results part (lines 157-159): “Almost 9 out of 10 patients (n=657, 89.6%) had received ≥1 dose of vaccine against SARS-CoV-2 with 550 (75.1%) reporting vaccination with ≥3 doses and the vast majority of the administered vaccines being mRNA-based (78.9-100% depending on the dose) (Table 1).”
Regarding the second part of your question (mRNA vaccines), please see our response in the previous comment (2 ii).
- Page 9, lines 215 – 234
Comment: The effect of age on BTI seems to have variable associations (positive and negative) based on the literature that the authors have referred to and may be related to different age groups that were studied in the current studies. Can the authors please highlight this in their discussion?
In the Discussion, we show the different association of age with severe COVID-19 (positive association) and breakthrough COVID-19 infection (negative association). Regarding the first, we refer you to the following part (lines 263-265): “As for severe BTI, older age, male sex, higher comorbidity index and specific comorbidities have been linked with increased risk17.” As for the second, we revised the following part (lines 284-287): “While increased age is correlated with severe BTI and similar to Coburn et al8 and studies from people without HIV16, younger PWH were more likely to develop non-severe BTI in our cohort, a finding possibly explained by the increased participation of those age groups in social gatherings or by the stricter adoption of social distancing and protective measures by older PWH with comorbidities.”
- Page 7, Figure 2A: regarding age
Comment: The Kaplan-Meier curves are shown for PWH under and over 60 years old. However, according to Table 2, taking that average and standard deviations inconsideration, none of the PWH were 60 or older than 60. It is unclear which data is used for the figure. Can the authors please clarify this in the text?
Eighty two out of 733 patients (11.2%) were older than 60 years. We have added a line in Table 1 where we now report these values.
- Page 7, Figure 2B: regarding prior COVID-19
Comment: Lines 158 – 160 (page 5) states that “In multivariable analysis, older age (per 1-year increase OR=0.97, 95% CI: 0.96- 158 0.99) and prior COVID-19 infection (OR=0.38, 95% CI: 0.21-0.68) were associated with lower risk for BTI…”. However, Figure 2B shows that incidence for breakthrough infections is high in younger (<60) PWH. It seems that the curves could have been labelled incorrectly. Can the authors please correct this? However, if the labels are correct, can the author please provide the source of the data for this figure, and discuss how this relates to their data and findings?
According to our analysis, a 1-year increase in age is associated with a 3% lower risk for breakthrough infection (BTI) (odds ratios below 1 are associated with reduced risk for the outcome). On the other way round, younger PWH have higher risk for BTI. This is correctly shown in the Figure 2A with those <60 years old having higher cumulative incidence of BTI, with this figure having been derived from our dataset by performing a Kaplan-Meier analysis followed by log-rank test implementation for the comparison of the BTI incidence among subgroups (<60 years vs >60 years old). We refer you to Figure 2 of reference #16 for a similar pattern of BTI incidence according to age.
- Page 5, lines 151 – 163 and Table 2 & 3: “Compared to those who 152 did not develop BTI after the booster dose, these participants were younger (44.2 ± 10.2 vs 153 47.4 ± 12 years, p=0.001)…” and “In multivariable analysis, older age (per 1-year increase OR=0.97, 95% CI: 0.96- 158 0.99) and prior COVID-19 infection (OR=0.38, 95% CI: 0.21-0.68) were associated with lower risk for BTI,…”
Comment: The age ranges between the BTI and no-BTI seem very close (44.2 vs. 47.4) but were associated with lower risk for BTI. In lines 215 – 234 (9), studies are cited that refer to “younger age” and “older age”. How “young” or “old” were these references referring to? Since difference in age is one of the main findings of this study, it needs to be presented better, especially since both the “younger” and “older” groups in this study are on average in the mid-40’s. Can the authors please present a figure or table that indicates the number of BTI and non-BTI per age group in 2- or 5-year intervals? This will clarify the “spread” in numbers of PWH in age intervals in the non-BTI and BTI to highlight the differences in age.
In response to your comment, we added a line in table 2 (Comparison of PLHIV who developed COVID-19 BTI after the booster dose vs those who did not), where we show the distribution of age in those with and without BTI. We chose 10-year intervals for two reasons. Firstly, for our results to be easily compared with the literature and secondly, to be easier to read. According to these results, it is clear that the percentage of patients >60 years was higher in those without vs those with BTI (15.4% vs 7.8%). It has to be noted, however, that the majority of our patients in both groups are between 40-60 years (57.1% and 59.1%, respectively). We address this issue in the Discussion by adding the following parts (lines 287-290 and 331-333): “It has to be noted, however, that our population was younger than those included in the aforementioned studies. For example, in the study by Coburn et al, 67% of PWH were older than 55 years, whereas the respective percentage in our study was 21.1%.” and “We also acknowledge that our study population was mainly young male PWH and for this reason our findings may not be easily extrapolated to more elderly or female patients.”
Minor concerns with the study:
- Page 2, line 72 - 73: “The participants were interviewed by the staff of the HIV Clinic (physicians and nurse).”
Suggested change: “The participants were interviewed by the staff of the HIV Clinic (physicians and a nurse).” Or “and nurses”.
Thank you, the suggested change was incorporated into the main text (lines 110-111).
- Page 2, line 86: “COVID-19 infection in participants at least 14 days after the 3rd vaccine (booster) dose..“
Suggested change: “COVID-19 infection in participants at least 14 days after the 3rd vaccine (booster) dose.“
Thank you, the suggested change was incorporated into the main text (lines 117-118).
- Page 6, Table 2: the text in the column headers are currently not visible. Please correct.
We changed the color of column headers in all tables in order for the text to be more visible.
- Page 7, Figure 1 legend: “Figure 1. Forest plot of multivariate Cox regression analysis of factors associated with BTI after 170 booster vaccine dose. Note: Odds ratios (ORs) lower than 1 are depicted in red and ORs higher than 1 in 171 blue. P values <0.001 are marked with (**) and those <0.001 with (***).”
Comment: ** and *** both refer to p<0.001 and p<0.001. Please correct.
The note of Figure 1 was corrected as follows (lines 207-208): “P values <0.01 are marked with (**) and those <0.001 with (***)”
- Page 8, lines 209 – 210: “not differ from the respective rates described by others during the same period of Omin-cron predominance.”
Suggested change: “not differ from the respective rates described by others during the same period of Omni-cron predominance.”
This part was corrected as follows (lines 255-256): “The reported peak incidence here does not differ from the respective rates described by others during the same period of Omicron predominance12.”
- Page 9, line 238: “transmission were associated with increased risk. We reaffirmed the substantial effective”
Suggested change: “transmission were associated with increased risk. We reaffirmed the findings of Tseng et al. that substantial effective”
We revised the text accordingly (lines 281-283): “We reaffirmed the findings of Tseng et al15 showing the substantial effectiveness of vaccination against severe COVID-19…”
- Page 10, lines 273 – 274: In addition, we collected and analyzed patient-level data, thus reducing the risk for noni-dentified confounding factors”
Suggested change: “Page 10, lines 273 – 274: In addition, we collected and analyzed patient-level data, thus reducing the risk for non-identified confounding factors”
This part was corrected as follows (lines 318-319): “In addition, we collected and analyzed patient-level data, thus reducing the risk for non-identified confounding factors.”
Reviewer 2 Report
Comments and Suggestions for Authors
The authors have analyzed the break through infection of COVID 19, Delta and Omicron strains in PLWH at a clinical facility in Greece.
The study could be an important data in the overall knowledge of COVID 19 vaccination status in HIV patients; however I have the following suggestions/comments:
The authors in their introduction mention "Limited data exist concerning breakthrough COVID-19 infections (BTI) in PWH." However a lot of study has already been conducted and reported on these lines. The authors need to modify this statement and should include the existing knowledge on this subject.
As mentioned above, the study lacks novelty.
The break through infection in healthy individuals from the same area should have been used as a control to actually corroborate the results. The control seems to be missing. The authors should come up with the explanation on this in the discussion section.
Did the authors have the idea, which strain of COVID 19 was responsible for the break through infection?
If other health conditions were taken into the account?
Most of the studied individuals are males. This also biases the overall study.
The title of the manuscript should be shortened.
Comments on the Quality of English LanguageMinor editing in the language would help in improving the clarity of the manuscript.
Author Response
Comments and Suggestions for Authors
The authors have analyzed the breakthrough infection of COVID 19, Delta and Omicron strains in PLWH at a clinical facility in Greece. The study could be an important data in the overall knowledge of COVID 19 vaccination status in HIV patients; however I have the following suggestions/comments:
- The authors in their introduction mention "Limited data exist concerning breakthrough COVID-19 infections (BTI) in PWH." However a lot of study has already been conducted and reported on these lines. The authors need to modify this statement and should include the existing knowledge on this subject.
Thank you, we revised this part as follows (lines 92-96): “While breakthrough COVID-19 infections (BTI) have been extensively described in the general population, relatively limited data exists concerning BTI in PWH especially during the Omicron period.”
We have also performed an additional search for relevant studies in the literature and we retrieved those of Yang et al (ref. #24) and of Sun et al (ref. #9). According to the study by Sun et al (ref. #9), we have revised the following part of the Introduction:
“Existing knowledge points to elevated rates of BTIs in immunocompromised patients, fully vaccinated PWH included, when compared to the general population, however this evidence derives from mainly from studies conducted during the pre-Omicron period8,9.” (lines 92-96)
The findings of the study by Yang et al (ref. #24) are discussed in the following parts:
“Similarly, the proportion of PWH over 60 years in the study of Yang et al was 28%24 with the respective rate in our study being 11%.” (lines 290-291)
“Yang et al also reported 79% decreased risk for BTI with an additional 3rd dose24.” (line )295
“We found a 62% decrease in the likelihood for BTI in PWH with a history of prior COVID-19, similar to the 63% decrease in BTI reported in a recent study of PWoH and PWH by Yang et al24.” (lines 300-302)
“The absence of a comparator group (PWoH) consists of another limitation of our study that precludes the generalization of our results regarding the incidence of BTI in the general population. Nevertheless, a recent study that compared the BTI rates between PWoH and PWH with high CD4 counts (66% with >500 cells/mm3) did not find any difference24.” (lines 333-336)
- As mentioned above, the study lacks novelty.
We believe that our study has two distinctive features that add important and valuable data to the existing public health literature. The first is that it spans until the end of 2022, including a substantial proportion of the Omicron predominance period, whereas other similar studies either do not include breakthrough infections during the Omicron period or include a shorter period of Omicron predominance (see ref. #28 by Yang et al). The second important feature is that, in contrast to other studies that study the characteristics of breakthrough infections after the initial two vaccine doses, we focused on breakthrough infections after an additional third booster dose. Given this, we believe that our study is more relevant and closer to the real-world status of COVID-19 vaccination, where most patients with risk factors have received at least one booster dose. This comment was added in the strengths of our study (lines 319-323).
- The breakthrough infection in healthy individuals from the same area should have been used as a control to actually corroborate the results. The control seems to be missing. The authors should come up with the explanation on this in the discussion section.
We agree that a control group of individuals without HIV infection would add substantially to the study. Nevertheless, such data were not available. We discuss this limitation in the Discussion section as follows (lines 333-335): “The absence of a comparator group (PWoH) consists of another limitation of our study that precludes the generalization of our results regarding the incidence of BTI in the general population. Nevertheless, a recent study that compared the BTI rates between PWoH and PWH with high CD4 counts (66% with >500 cells/mm3) did not find any difference24.”
- Did the authors have the idea which strain of COVID 19 was responsible for the breakthrough infection?
We mention that the majority of the BTI occurred during the predominance of Omicron variant (lines 170-173): “Of note, the epidemic curves for the first COVID-19 infection and first BTI after the booster dose mirrored those of the general population in Greece, with the highest incidence rates observed in June 2022 (during Omicron variant predominance)”. We also refer you to Suppl. Figure 2B, where we show that the both peaks of BTI (Feb-March 2022 and Jun-July 2022) occurred during almost exclusive Omicron strain circulation in Greece.
- If other health conditions were taken into the account?
Several comorbidities were recorded and included in the univariable analysis, without showing significant association with breakthrough COVID-19 infection (BTI) (Table 2), with the exception of body mass index that was finally included in the multivariate logistic regression (Table 3). We believe that we have recorded most of the clinically significant health conditions that could affect the outcome (BTI). We, however, now recognize in the Discussion that “In addition to this, the presence of other non-identified health conditions that could affect the outcome, cannot be excluded.” (lines 330-331)
- Most of the studied individuals are males. This also biases the overall study.
Thank you for this comment. We have the added this as a limitation in the Discussion (lines 331-333): “We also acknowledge that our study population was mainly young male PWH and for this reason our findings may not be easily extrapolated to more elderly or female patients.”
- The title of the manuscript should be shortened.
We revised the title as follows: “Breakthrough COVID-19 infections after booster SARS-CoV-2 vaccination in a Greek cohort of people living with HIV during Delta and Omicron wave”
- Comments on the Quality of English Language: Minor editing in the language would help in improving the clarity of the manuscript.
Thank you for the valuable comment, we tried to edit the text accordingly.
Round 2
Reviewer 2 Report
Comments and Suggestions for Authors
The authors have addressed my concerns and I have no further comments.